# Identifying Cardiovascular Risk Profiles Clusters among Mediterranean Adolescents across Seven Countries

**DOI:** 10.3390/healthcare10020268

**Published:** 2022-01-29

**Authors:** Riki Tesler, Sharon Barak, Orna Reges, Concepción Moreno-Maldonado, Rotem Maor, Tânia Gaspar, Oya Ercan, Yael Sela, Gizell Green, Avi Zigdon, Adilson Marques, Kwok Ng, Yossi Harel-Fisch

**Affiliations:** 1Department of Health Systems Management, School of Health Sciences, Ariel University, Ariel 4076405, Israel; orna.reges@gmail.com (O.R.); aviz@ariel.ac.il (A.Z.); 2Program in Gerontology, Faculty of Health Sciences, Ben-Gurion University of the Negev, Beer Sheeva 8855630, Israel; sharoni.baraki@gmail.com; 3Department of Developmental Psychology, University of Sevilla, 41004 Sevilla, Spain; morenomaldonado.mc@gmail.com; 4School of Education, Bar Ilan University, Ramat Gan 5290002, Israel; rotemmaor@hotmail.com (R.M.); harelyossi@gmail.com (Y.H.-F.); 5Sleep Medicine Center—CENC, 1070-068 Lisbon, Portugal; Tania.Gaspar-Barra@hbsc.org; 6Instituto de Saúde Ambiental (ISAMB), Faculdade de Medicina, Universidade de Lisboa, 1649-028 Lisbon, Portugal; 7Comprehensive Health Research Center (CHRC), Nova Medical School, Universidade Nova de Lisboa, 1160-056 Lisbon, Portugal; 8Divisions of Pediatric Endocrinology and Adolescent, Faculty of Medicine, Istanbul University-Cerrahpasa, Istanbul 34452, Turkey; oyaercan@istanbul.edu.tr; 9Faculty of Social & Community Sciences, Ruppin Academic Center, Emek Heffer 4025000, Israel; yaelvile@gmail.com; 10Department of Nursing, Faculty of Health Science, Ariel University, Ariel 4076405, Israel; ggrin@campus.haifa.ac.il; 11CIPER, Faculdade de Motricidade Humana, Universidade de Lisboa, 1499-002 Lisbon, Portugal; amarques@fmh.ulisboa.pt; 12Physical Activity for Health Research Cluster, Department of Physical Education and Sport Sciences, University of Limerick, V94 T9PX Limerick, Ireland; kwok.ng@hbsc.org; 13School of Educational Sciences and Psychology, University of Eastern Finland, 80100 Joensuu, Finland

**Keywords:** lifestyle behavior, cardiovascular risk factors, youth, cluster analysis

## Abstract

Cardiovascular diseases (CVDs) are the number one cause of death globally and are partially due to the inability to control modifiable lifestyle risk factors. The aim of this study was to analyze the profiles of adolescents from seven Mediterranean countries (Greece, Israel, Italy, Macedonia, Malta, Portugal, Spain) according to their modifiable lifestyle risk factors for CVD (overweight/obesity, physical activity, smoking, alcohol consumption). The sample consisted of 26,110 adolescents (52.3% girls) aged 11, 13, and 15 years who participated in the Health Behavior in School-aged Children (HBSC) survey in 2018 across the seven countries. Sociodemographic characteristics (sex, age, country of residence, socioeconomic status) and CVD modifiable lifestyle risk factors (overweight/obesity, physical activity, smoking, alcohol consumption) were recorded. A two-step cluster analysis, one-way analysis of variance, and chi-square test were performed. Four different cluster groups were identified: two low-risk groups (64.46%), with risk among those with low physical activity levels; moderate-risk group (14.83%), with two risk factors (unhealthy weight and low physical activity level); and a high-risk group (20.7%), which presented risk in all modifiable lifestyle risk factors. Older adolescents reported a higher likelihood of being in the high-risk group. Given that the adolescence period constitutes an important time for interventions aimed at CVD prevention, identifying profiles of moderate- and high-risk adolescents is crucial.

## 1. Introduction

Cardiovascular disease (CVD) is a term that encompasses various heart and blood vessel disorders mostly caused by atherosclerosis (e.g., hypertension, coronary heart disease, congestive heart failure, stroke, atrial fibrillation) [1,2]. CVD can evolve throughout one’s lifetime, often displaying no symptoms. Symptoms can lead to severe deterioration or may manifest as sudden death [3]. Approximately 60% of all cardiovascular-related deaths occur among asymptomatic people who had not previously suffered a cardiovascular incident [4]. CVDs remain the leading cause of death globally [5]. CVD cases have nearly doubled from 271 million in 1990 to 523 million in 2019. Furthermore, the number of CVD deaths has increased from 12.1 million in 1990 to 17.9 million lives in 2021 [6]. By 2030, the annual rate of premature CVD-related deaths is expected to reach roughly 24 million worldwide [7].

The high prevalence of CVD may be attributed to the inability to control the disease’s related risk factors [8]. There are four types of CVD risk factors, namely non-modifiable (e.g., age, sex, family history of CVD); modifiable (e.g., blood pressure, total cholesterol, blood sugar); social (e.g., income, social deprivation, environment); and lifestyle (e.g., smoking, diet, physical inactivity) [9]. Preventive intervention for CVD is widely recognized as an effective way to reduce the incidence and progression of these risk factors [10,11,12,13]. Lifestyle factors have generally been the main focus of attention in health education and promotion [10,13,14,15].

Although the clinical appearance of CVDs usually takes place during adulthood, the majority of its precursors are common among adolescents [16], with signs of these risk behaviors appearing throughout adolescence [17]. While behaviors implemented during one’s developmental years tend to continue into maturity [17,18], clustering these risk factors among adolescents may hasten atherogenesis [19,20]. A recent study demonstrated that poor diet factors, such as low fruit intake, were present in more than half of the adolescent population studied [21,22,23]. Moreover, risk factors tend to appear together [17,22,24]. Therefore, there is a need to estimate risk behaviors (physical inactivity, smoking, alcohol consumption, and overweight/obesity) of CVD and examine their clustering patterns among adolescents in various countries.

Identifying clusters of modifiable risk factors allow for the design of health-oriented intervention programs of health promotion during adolescents’ formative years. In addition, it is believed that the relationship between CVD and its risk factors depends on ethnicity and geographical region, which underscores the need for an investigation into these factors [25,26]. Accordingly, the aim of this study was to analyze adolescent profiles from seven Mediterranean countries (Greece, Israel, Italy, Macedonia, Malta, Portugal, Spain) according to their modifiable lifestyle CVD risk factors (overweight/obesity, physical activity, smoking, alcohol consumption). The study focuses only on Mediterranean countries, as the Mediterranean Region has many unique geographical, morphologic, historical, and societal characteristics related to CVD risk factors. For example, a growing body of evidence has shown that the traditional Mediterranean diet is associated with lower rates of both chronic diseases and premature death [27].

## 2. Materials and Methods

### 2.1. Study Setting

Data for this study were obtained from the Health Behavior in School-aged Children (HBSC) study, a cross-national survey with support from the World Health Organization (WHO, Europe), which was completed in 2018/2019 in 47 countries, including 40 European countries, Canada, and Israel. More detailed background information about the study is provided on HBSC’s website and in international reports [28].

### 2.2. Research Design

The current study included adolescents from the HBSC study from following seven different Mediterranean countries: Greece, Israel, Macedonia, Malta, Spain, Italy, and Portugal aged 11, 13, and 15 years. Sampling was conducted in accordance with the structure of national education systems within each country. In most countries, the primary sampling unit was the school class or the whole school when a sample class was not available. If a school with two or more classes was selected, the one chosen for the sample was randomly selected [29].

### 2.3. Data Collection and Survey Instrument

Data were collected through self-reported standardized questionnaires. The surveys were administrated in school classrooms. Students did not provide any personal details (e.g., name, classroom, teacher), making them completely anonymous and ensuring their confidentiality. Researchers strictly followed the standardized international research protocol to ensure consistency in survey instruments, data collection, and processing procedures [29,30]. Response rates at the school, class, and student level exceed 80% in most countries [29].

The HBSC included items describing the children’s sociodemographic characteristics. More specifically, children reported their self-identified sex (boy, girl, and in some countries “neither term describes me”), age, country of residency (Greece, Israel, Macedonia, Malta, Spain, Italy, or Portugal), and socioeconomic status (using the revised Family Affluence Scale III [FAS]). The FAS is a 6-item measure of material assets in the home, measured based on the number of vehicles owned, bedroom sharing, computer ownership, number of bathrooms at home, dishwashers at home, and number of family vacations taken [31]. The scale has a total score that ranges from 0 (lowest) to 17 (highest). The FAS has been shown to be a better proxy of socioeconomic status than measures that rely on adolescent reports of parental occupation or income [32].

### 2.4. Measures and Insatruments

The four following different types of modifiable lifestyle characteristics were examined: obesity, physical activity, smoking, and alcohol consumption. Below, we describe each of these aforementioned outcomes.

#### 2.4.1. Overweight and Obesity

Overweight and obesity were measured according to students’ self-reported weight and height, which were then calculated into body mass index values (kilogram per square meter). Bodyweight status was assessed according to the International Obesity Task Force’s cut-off values [33], divided into three categories: underweight/normal weight, overweight, and obese. In the current study, students’ weight status was categorized into “overweight or obese” or “not overweight or obese”.

#### 2.4.2. Physical Activity Habits: Moderate Physical Activity

According to a previously published study, physical activity was defined as “any activity that usually increases your heart rate and makes you get out of breath some of the time” [34]. Accordingly, physical activity level was assessed using the question “How often over the past seven days have you been physically active for a total of at least 60 min per day?” Answers were given on an 8-point scale (0 = none to 7 = daily). The measure has reasonable validity (r = 0.37) with 5-day accelerometer data [35] and acceptable test-retest reliability when used as a dichotomous variable [36]. At the time of data collection, according to the WHO (2021), adolescents and young people aged 5–17 years should engage in at least 60 min of moderate physical activity daily [37]. The mean number of active days was calculated and used in the cluster analysis. In addition, participants’ responses were converted into the following two categories: not meeting and meeting the WHO recommendations; the prevalence for each group was calculated.

#### 2.4.3. Physical Activity Habits: Vigorous Physical Activity

Engagement in vigorous physical activity out of school was also assessed according to the WHO’s recommendations, which is that children and youth should engage in vigorous physical activity at least three days a week. More specifically, participants were asked: “After school, how often do you usually engage in your free time in physical activity that causes you to heavily breathe or sweat?” The measurement scale was (1) every day; (2) 4–6 times/week; (3) 2–3 times/week; (4) once a week; (5) once a month; (6) less than once a month; and (7) never. Accordingly, participants were grouped into those who met (answers 1 through 3) and did not meet this recommendation (answers 4 through 7) [37].

#### 2.4.4. Smoking Habits

In the current study, the question investigates the frequency of tobacco-smoking habits (“How often did you smoke tobacco in the past 30 days?”), with response options ranging from “I do not smoke” to “every day”. The authors of Charrier et al. (2019) [38] report how this was used for the analysis (cut-off or continuous).

#### 2.4.5. Alcohol Consumption Habits

In the current study, the frequency of alcohol consumption was investigated by asking, “At what frequency do you drink alcohol (beer, wine/sparkling wine, or spirits/liquor)?” The following multiple-choice answers were included: (a) every day; (b) every week; (c) every month; (d) rarely; and (e) never used. The respondents were assigned to the following groups according to the frequency of alcohol consumption: (a) not users of alcohol (answered “never used” for all three categories of alcoholic beverages); irregular users (drank anything alcoholic “every month” or less frequently); or regular users (used beer, wine/sparkling wine, or spirits/liquor “every week” or more often).

Drunkenness was assessed with the question “In the past 30 days, have you ever had so much alcohol that you were really drunk?” The following multiple-choice answers were offered: (a) no, never; (b) yes, once; (c) yes, 2–3 times; (d) yes, 4–10 times; and (e) yes, more than 10 times. The proportions of those who claimed to have never been drunk, having been drunk once, and having been drunk twice or more are presented as cut-off points. Self-reports on drunkenness provide a measure of excessive alcohol use [39].

### 2.5. Data Analysis

Descriptive statistics (means, standard deviations, sample sizes, and percentages) were used to describe the sociodemographic and lifestyle characteristics of the total sample. Six variables assessing four different modifiable lifestyle cardiovascular risk factors were included in the cluster analysis as follows: obesity (overweight/obese vs. not overweight or obese), physical activity level (number of days conducting moderate physical activity level and meeting/not meeting the recommendations for vigorous physical activity), smoking habits (number of cigarettes smoked/week), and drinking habits (regular user/irregular user/not user and number of days drunk in the past month). We conducted a two-step cluster analysis. This type of analysis is a hybrid approach that first uses a distance measure to separate groups and then a probabilistic approach to choose the optimal subgroup model [40]. This type of cluster analysis has numerous advantages compared to more cluster analysis traditional techniques, as it can determine the number of clusters based on a statistical measure of fit rather than on an arbitrary choice by using categorical and continuous variables simultaneously, analyzing outliers, and being able to handle large datasets [40,41]. Two-step cluster analysis is considered reliable in terms of the number of subgroups detected, classification probability of individuals to subgroups, and reproducibility of findings on clinical and other types of data [40]. In the first step (pre-clustering), a sequential approach was used to pre-cluster the cases based on the definition of dense regions in the analyzed attribute space. Then, the pre-clusters were statistically merged in a stepwise manner until all clusters were in one cluster [42]. The cut-off for favorable lifestyle behavior was set at 20%, meaning < 20% of the sample presenting unfavorable lifestyle behaviors. Based on the number of unfavorable risk factors in each cluster, clusters were labeled as “low risk” (zero or one unfavorable behavior), “moderate risk” (2 unfavorable behaviors), and “high risk” (>3 unfavorable behaviors). Finally, the clusters’ sociodemographic characteristics (age, socioeconomic status, sex, and country) were evaluated and compared. The continuous variables (age and socioeconomic status) were evaluated using one-way analysis of variance with Tukey–Kramer post-hoc test. Normal distribution was also evaluated using the D’Agostino–Pearson test. Categorical variables were evaluated using chi-square test. In all analyses, the level of significance was set to *p* < 0.05 (two-tailed) using IBM SPSS Statistics (version 23.0, IBM, Armonk, NY, USA).

## 3. Results

### 3.1. Survey Findings: Sociodemographic Characteristics of the Total Sample

The total study population (*n* = 26,110) was composed of 13,656 females and 12,454 males (mean age: 13.57 + 1.61). Participants were recruited from seven different Mediterranean countries, with the highest prevalence of participants from Israel (23.09% of the sample) and Portugal (18.4% of the sample) and the smallest from Malta (7.69% of the sample; Table 1).

### 3.2. Modifiable Lifestyle Characteristics of the Total Sample

Four different modifiable lifestyle characteristics were examined: obesity, physical activity level, smoking habits, and drinking habits. The results suggested that although most of the study participants had a healthy weight, approximately 20% of the sample was overweight or obese. Regarding physical activity level, the mean number of days in which participants accumulated 60 min of moderate physical activity/day was 3.72 days/week + 2.13. Moreover, 16% of the sample reached the recommended level of moderate physical activity level. Vigorous physical activity was conducted sufficiently by approximately 70% of the sample. Most participants reported that they did not smoke or drink in the past 30 days (93% and 80% of the sample, respectively). Similarly, most participants were not drunk in the past 30 days (95% of the sample; Table 2). In addition, among the 26,110 participants, approximately 15% had zero healthy lifestyle behaviors, while most presented one healthy lifestyle behavior (49.6%).

### 3.3. Cluster Analysis

The two-step cluster analysis reported a four-cluster classification as the optimal solution for the data considered in the present study. Following a parsimony criterion, the four-cluster solution presented the greatest ratio distance measure (1.45), which is based on the current number of clusters against the previous number of clusters.

The four clusters varied in terms of cardiovascular risk factor prevalence. According to the risk stratification labeling, there were two low-risk clusters. Children in the first low-risk cluster (*n* = 8115, 31.08%) did not present any cardiovascular risk behaviors except for not engaging sufficiently in daily physical activity. Similarly, the second low-risk cluster (*n* = 8718, 33.38%) also presented only a low physical activity level. The third cluster (*n* = 3874, 14.83%) presented a moderate risk for CVD. The unhealthy lifestyles in this cluster included unhealthy weight and low levels of physical activity. Finally, the fourth cluster (*n* = 5403, 20.69%) was composed of children who presented a high risk for CVD, which included children presenting unfavorable behaviors in all risk factors evaluated (Table 3; Figure 1).

### 3.4. Sociodemographic Characteristics of the Four Clusters

Significant between-cluster differences were found regarding participants’ age. More specifically, participants’ ages in clusters 1 (low-risk—group 1), 2 (low-risk—group 2), and 3 (moderate-risk) were significantly younger than that of cluster 4 (high-risk; 13.14 + 1.52, 13.27 + 1.53, 13.47 + 1.5, and 14.74 + 1.31, respectively; F-ratio = 1385.37; *p* < 0.0001). In addition, the socio-economic status of clusters 1 and 2 (low-risk-groups 1 and 2) were significantly higher than that of clusters 3 (moderate-risk) and 4 (high-risk; 8.30 + 2.43, 8.50 + 2.43, 7.75 + 2.45, 7.94 + 1.31; F-ratio = 109.65; *p* < 0.001).

The highest prevalence was found in cluster 1 (low-risk—group 1) among males and females. However, 18% of the females belonged to cluster 4 (high-risk) in comparison to 23% of the males (Figure 2).

**Figure 1 healthcare-10-00268-f001:**
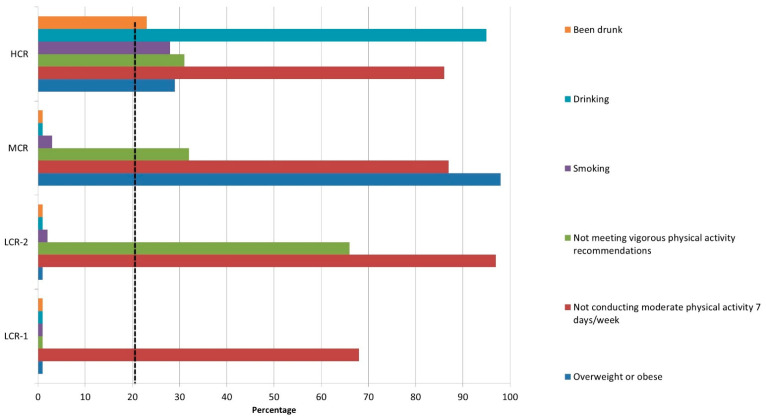
Within-cluster percentage of favorable lifestyle behavior. Abbreviations: HCR, high cardiovascular risk; LCR-1, low cardiovascular risk—group 1; LCR-2, low cardiovascular risk—group 2; MCR, moderate cardiovascular risk. Notes: The vertical dashed black line represents the cut-off for favorable lifestyle (<20% of the sample presented unfavorable lifestyle behavior); The dotted line represents the cut-off for favorable lifestyle behavior, meaning <20% of the sample presenting unfavorable lifestyle behaviors.

In all seven countries, the most prevalent clusters were low-risk groups 1 and 2 (Figure 3). However, the prevalence of combined clusters 1 and 2 (the two low-risk clusters) varied and ranged from 54% (Malta) to 73% (Israel). The prevalence of cluster 4 (high-risk) ranged from 14% (Israel) to 31% (Greece).

## 4. Discussion

To the best of our knowledge, despite the wide use of cluster analyses in the literature, no study to date has compared CVD lifestyle risk factors among adolescents from various Mediterranean countries. Considering the four CVD lifestyle risk factors assessed, the best clustering solution, which also yielded meaningful data taxonomy, was four. As CVD is the leading cause of death globally [26], early detection of lifestyle risk factors and their clustering among adolescents may be crucial in managing, controlling, and preventing future CVD.

The prevalence of the four cardiovascular risk factors assessed was very distinct. The most prevalent lifestyle risk factor was low physical activity level. This risk factor was present in all the clusters, including in the low-risk clusters. These results were not surprising, as high rates of physical inactivity at young ages have been found worldwide [35,36,37]. For example, in the National Health and Nutrition Examination Survey, roughly 25% of adolescents met the physical activity guidelines of 60 min of moderate to vigorous physical activity daily [42]. These results are highly concerning, as regular physical activity can not only reduce a person’s risk of dying from CVD [24,26] but is also highly important for adolescent’s physical (e.g., aid in forming strong bone development) and mental (e.g., better quality of life) health [43]. Therefore, increasing youth physical activity levels is cardinal and should be conducted at the national level [44]. The reduction of physical education in schools is problematic. One of the prevalent physical activity barriers that youth face is lack of time [45]. It is also important to understand facilitators to physical activity. For example, the two most popular facilitators to physical activity are self-efficacy and beliefs of the benefits of physical activity [46]. Accordingly, understanding possible differences in physical activity barriers/facilitators between youth and different countries would enable those designing health promotion interventions to develop more specific and effective interventions to increase physical activity among youth.

In the study, older adolescents and those from lower socioeconomic status reported a higher likelihood of belonging to the high-risk group. The increased likelihood of being in the high-risk group among older participants was not surprising, as it is known that risky behaviors increase with age. For example, according to the Centers for Disease Control and Prevention (2017), the prevalence of drinking increased from 23.4% among 9th grade students to 42.4% among 12th grade students. Similarly, the prevalence of binge drinking was 10.4% among 9th grade students and 24.6% among 12th grade students. Moreover, physical activity, the most prevalent lifestyle risk factor, also tends to decrease with age. Accordingly, adolescents constitute one of the most sedentary populations [47]. Additionally, socioeconomic status plays an important role in adopting healthy lifestyle behaviors. For example, numerous studies have shown that adults from lower socioeconomic status are more likely to smoke. However, the literature addressing smoking among youth is inconsistent. Several studies have shown an association between smoking and socioeconomic status among youth in some European countries, while others have not uncovered such associations [48]. In the absence of scientific agreement regarding the role of socioeconomic status and healthy lifestyle behaviors, such as smoking, among youth, the current study indicates the need for specific policies and interventions for different target groups in order to increase healthy lifestyle behaviors associated with reduced CVD.

This study had several limitations. First, this was a cross-sectional study; it only provides an association and not causation. Second, the application of convenience sampling and recalled method to collect data may elevate the risk of selection bias and recall bias, respectively. Although we adjusted all the potential risk factors in the binary logistic regression, there is also a possibility that there were residual confounders. Besides these, blood pressure measurement is not included in this study, which is an important CVD risk factor. Another limitation concerns the comparability of data across countries. Constructs such as health complaints may be understood differently in different settings or contexts. For some indicators, such as subjective health complaints, cross-national comparability has already been tested but not in all countries for factual questions (e.g., height and weight) and behavior reports; there may be cultural differences regarding socially desirability response sets [29]. However, because all indicators have been dichotomized, slight exaggerations and understatements should not play a role.

This study also has many strengths. For example, we address, for the first time, the prevalence of CVD risk factors among school children in Mediterranean countries. In public health, it may be beneficial for Mediterranean countries to collaborate to develop and implement CVD-prevention programs. Identifying profiles of moderate- and high-risk adolescents is crucial.

It is well acknowledged that the connections between lifestyle behaviors, such as physical inactivity, unhealthy eating, smoking, and excessive alcohol consumption, and CVD can be modified. Much change can be initiated through lifestyle interventions although some prescriptions are available for adolescents to use [49]. Adolescent smoking is common among Mediterranean countries, and in both the medium- and high-risk groups, there were regular smokers (International Report), suggesting further work is needed to establish ways to reduce this risk [22,29]. The high-risk group consists of regular smoking, drinking, and physical inactivity, suggesting a set of concomitant behaviors that is multifactorial, mimicking the possibilities for adulthood behavior. Peers play an important role into smoking and alcohol consumption, particularly drunkenness. During adolescence, their drive for thrill-seeking behaviors is an integral part of the maturation process. However, this should not be used as an excuse to normalize such reckless behaviors [8,50]. Participants in regular sports in a healthy environment can also be a good mechanism to reduce CVD risks although this also requires sport clubs to take the responsibilities to produce health-enhancing physical activity and sport [2,8,9,10].

## 5. Conclusions

To the best of our knowledge, our study is the first conducted within and between adolescents from Mediterranean countries that uncovers different CVD health risks. The study identified four different cluster groups for CVD risk factors. In addition, older adolescents and those from lower socioeconomic status reported a higher likelihood of belonging to the high-risk group. Given the growing rate of premature CVD-related deaths worldwide, public health authorities and governments should consider implementing policies, health promotion, and tailored preventive strategies to increase healthy lifestyle behaviors associated with reduced CVD. CVD-risk clustering is an important step in strategy development, as it may identify the most high-risk population for CVD. For example, the study’s results show that youth from lower SES families are more likely to present CVD-risk factors, suggesting that the particular needs of this population should be addressed and programs developed to increase healthy lifestyle behaviors.

## Figures and Tables

**Figure 2 healthcare-10-00268-f002:**
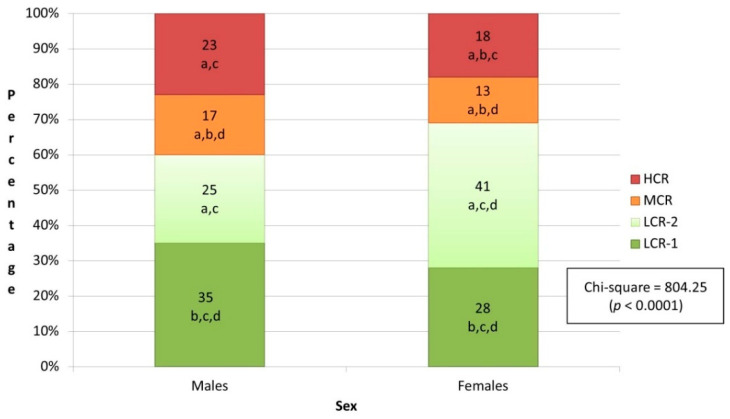
Clusters’ distribution based on sex. Notes: Abbreviations: HCR, high cardiovascular risk; MCR, moderate cardiovascular risk; LCR-1, low cardiovascular risk—group 1; LCR-2, low cardiovascular risk—group 2. Notes: a, statistically significantly different from first cluster (*p* < 0.05; 2-tailed); b, statistically significantly different from second cluster (*p* < 0.05; 2-tailed); c, statistically significantly different from third cluster (*p* < 0.05; 2-tailed); d, statistically significantly different from fourth cluster (*p* < 0.05; 2-tailed).

**Figure 3 healthcare-10-00268-f003:**
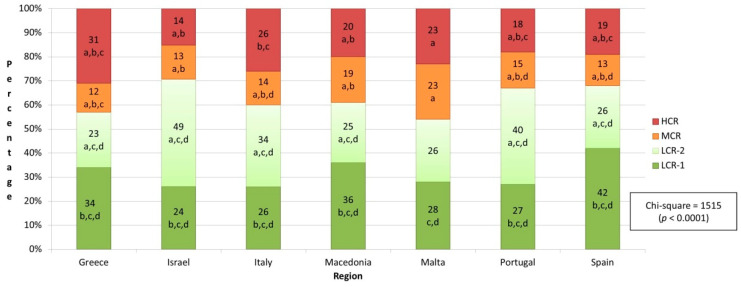
Clusters’ distribution based on region. Abbreviations: HCR, high cardiovascular risk; MCR—moderate cardiovascular risk; LCR-1, low cardiovascular risk—group 1; LCR-2, low cardiovascular risk—group 2; Notes: a, statistically significantly different from first cluster (*p* < 0.05; 2-tailed); b, statistically significantly different from second cluster (*p* < 0.05; 2-tailed); c, statistically significantly different from third cluster (*p* < 0.05; 2-tailed); d, statistically significantly different from fourth cluster (*p* < 0.05; 2-tailed).

**Table 1 healthcare-10-00268-t001:** Sociodemographic characteristics of the study sample (*n* = 26,110).

Variables		Mean (SD) (Range) OR *N* (%)
Age, years: mean (SD)		13.57 (1.61) (10.5–16.5)
Sex: *n* (%)	Male	12,454 (47.69)
	Female	13,656 (52.30)
Socioeconomic status (Family Affluence Scale): mean (SD) (range)		7.96 (2.49) (0.00–17.00)
Country: *n* (%)	Spain	3368 (12.89)
	Greece	3028 (11.59)
	Israel	6031 (23.09)
	Italy	3237 (12.39)
	Macedonia	3629 (13.89)
	Malta	2010 (7.69)
	Portugal	4807 (18.41)

Abbreviation: SD, standard deviation.

**Table 2 healthcare-10-00268-t002:** Lifestyle characteristics of the study sample (*n* = 26,110).

Variable			Mean (SD) (Range) OR *N* (%)
Weight	Overweight and obese: *n* (%)	Yes	5004 (19.16)
		No	21,106 (80.83)
Physical activity habits	60 min of moderate physical activity/day: mean (SD) (range)		3.72 (2.13) (0.00–7.00)
	Vigorous physical activity 3x week: *n* (%)	Not meeting recommendations	7968 (30.51)
		Meeting recommendations	18,142 (69.48)
Smoking habits	Days smoking in the past 30 days: *n* (%)	Every day	432 (1.65)
		At least once a week but not every day	430 (1.64)
		Less than once a week	852 (3.26)
		No smoking	24,396 (93.43)
Drinking habits	Drinking alcohol status: *n* (%)	Regular users	1907 (7.30)
		Irregular users	3186 (12.20)
		Not users	21,017 (80.49)
	Days drunk in the past 30 days: *n* (%)	Twice or more	378 (1.44)
		Once	835 (3.19)
		Never	24,897 (95.35)

Abbreviations: SD, standard deviations.

**Table 3 healthcare-10-00268-t003:** Description of the four clusters according to lifestyle behavioral characteristics (*n* = 26,110).

Variables			Cluster 1 (*n* = 8115): Mean (SD) (Range) OR *N* (%)	Cluster 2 (*n* = 8718): Mean (SD) (Range) OR *N* (%)	Cluster 3 (*n* = 3874): Mean (SD) (Range) OR *N* (%)	Cluster 4 (*n* = 5403): Mean (SD) (Range) OR *N* (%)	Chi-Square Test (*p*-Value) OR ANOVA Test (*p*-Value)
			Low Cardiovascular Risk—Group 1	Low Cardiovascular Risk—Group 2	Moderate Cardiovascular Risk	High Cardiovascular Risk	
Weight	Overweight and obese: *n* (%)	Yes	0 (0) ^c,d^	0 (0) ^c,d^	3874 (100) ^a,b,d^	1513 (28.00) ^a,b,c^	20,341 (<0.001)
		No	8115 (100) ^c,d^	8718 (100) ^c,d^	0 (0) ^a,b,d^	3890 (71.99) ^a,b,c^	
Physical activity	60 min of moderate physical activity/day: mean (SD) (range)		5.45 (1.18) ^b,c,d^ (4–7)	2.38 (1.50) ^a,c,d^ (0–7)	3.59 (2.04) ^a,b,d^ (0–7)	3.69 (2.10) ^a,b,c^ (0–7)	5183 (<0.001)
	Vigorous physical activity 3/week *n* (%)	Not meeting recommendations	0 (0) ^b,c,d^	5080 (58.27) ^a,c,d^	1239 (31.98) ^a,b^	1649 (30.52) ^a,b^	12.24 (<0.001)
		Meeting recommendations	8115 (100) ^b,c,d^	3638 (30.21) ^a,c,d^	2635 (68.01) ^a,b^	3754 (69.47) ^a,b^	
Smoking habits	Days smoking in the past 30 days: *n* (%)	Every day	0 (0) ^d^	0 (0) ^d^	24 (0.61) ^d^	408 (7.55) ^a,b,c^	4715 (<0.001)
		At least once a week but not every day	22 (0.27) ^d^	29 (0.33) ^d^	25 (0.64) ^d^	354 (6.55) ^a,b,c^	
		Less than once a week	12 (0.14) ^d^	86 (0.98) ^d^	56 (1.44) ^d^	698 (12.91) ^a,b,c^	
		No smoking	8081 (99.58) ^d^	8603 (98.68) ^d^	3769 (97.28) ^d^	3943 (72.97) ^a,b,c^	
Drinking habits	Drinking alcohol status: *n* (%)	Regular users	0 (0) ^d^	(0) ^d^	0 (0) ^d^	1907 (35.29) ^a,b,c^	4841.29 (<0.001)
		Irregular users	0 (0) ^d^	0 (0) ^d^		3186 (58.9) ^a,b,c^	
		Not users	8115 (100) ^d^	8718 (100) ^d^	0 (0) ^d^	310 (5.73) ^a,b,c^	
	Days drunk in the past 30 days: *n* (%)	Twice or more	0 (0) ^d^	0 (0) ^d^	0 (0) ^d^	378 (6.99) ^a,b,c^	1532.45 (<0.001)
		Once	0 (0) ^d^	0 (0) ^d^	0 (0) ^d^	835 (15.45) ^a,b,c^	
		Never	8115 (100) ^d^	8718 (100) ^d^	3874 (100) ^d^	4190 (77.54) ^a,b,c^	

Notes: ^a^ Statistically significantly different from first cluster (*p* < 0.05; 2-tailed); ^b^ statistically significantly different from second cluster (*p* < 0.05; 2-tailed); ^c^ statistically significantly different from third cluster (*p* < 0.05; 2-tailed); ^d^ statistically significantly different from fourth cluster (*p* < 0.05; 2-tailed); Abbreviation: SD, standard deviation.

## Data Availability

The data presented in this study are openly available in http://doi.org/10.1016/j.jadohealth.2020.02.012 (accessed on 12 February 2020).

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
