# Peer review of "Identifying Cardiovascular Risk Profiles Clusters among Mediterranean Adolescents across Seven Countries"

_healthcare, 2022, doi:10.3390/healthcare10020268_

Round 1

Reviewer 1 Report

General comments:

In their study: “Identifying Cardiovascular Risk Profiles Clusters Among Mediterranean Adolescents Across Seven Countries” Tesler R. et al. used cross-sectional data from more than 26,000 adolescent participants of the Health Behavior in School-aged Children (HBSC) survey across seven Mediterranean countries to identify prevalent cluster of cardiovascular disease (CVD) risk factors. Please find below detailed comments raised during review of the manuscript:

Major comments

Results: suggest to present a visualization of an index variable for “number of healthy lifestyle characteristics” (see Figure 1 in: Ford et al. Arch Intern Med. 2009;169(15):1355-1362).

Methods: please specify if there is a published full analysis of all 47 countries of the Health Behavior in School-aged Children (HBSC) survey, including 40 European countries, Canada, and Israel. If yes, what is a value of the present subgroup analysis among the Mediterranean countries of the HBSC? Please comment to prevent potential salami slicing/fragmented publication of full data sets (see doi: 10.5489/cuaj.11265).

Discussion, l.287-288: “The cluster analyses showed that the four separate clusters provide the most efficient CVD lifestyle risk factors profiling.” This statement is speculative, given that overall, only four CVD lifestyle risk factor were initially included in the data analysis without any forward or backward selection of a larger variable set.

Minor comments

Abstract: specify the number of modifiable CVD lifestyle characteristics that were examined: l. 33: “according to four modifiable lifestyle risk factors for CVD (…).”

Discussion, l: 291: “The prevalence of the five cardiovascular risk factors assessed.” Only four CVD risk factors were assessed – please correct.

Author Response

Response to Reviewer 1 Comments

General comments:

In their study: “Identifying Cardiovascular Risk Profiles Clusters Among Mediterranean Adolescents Across Seven Countries” Tesler R. et al. used cross-sectional data from more than 26,000 adolescent participants of the Health Behavior in School-aged Children (HBSC) survey across seven Mediterranean countries to identify prevalent cluster of cardiovascular disease (CVD) risk factors. Please find below detailed comments raised during review of the manuscript:

Point 1: Results: suggest to present a visualization of an index variable for “number of healthy lifestyle characteristics” (see Figure 1 in: Ford et al. Arch Intern Med. 2009;169(15):1355-1362).

Response 1: Thank you for your comment. We added more information in the "Results" section, lines 252-253

Point 2: Methods: please specify if there is a published full analysis of all 47 countries of the Health Behavior in School-aged Children (HBSC) survey, including 40 European countries, Canada, and Israel. If yes, what is a value of the present subgroup analysis among the Mediterranean countries of the HBSC? Please comment to prevent potential salami slicing/fragmented publication of full data sets (see doi: 10.5489/cuaj.11265).

Response 2:
Thanks for the comment. There are published data from all the countries that are part of the HBSC. These data are periodically released in the international HBCS reports. However, the analyses presented in this manuscript have not been previously performed. Furthermore, as Mediterranean countries have social, economic, cultural, and climatic characteristics significantly different from other HBSC countries, analyses only including these countries will give us regional-specific insights. Therefore, it is crucial to consider the similarities and differences between countries when performing subgroup analyses. Accordingly, we addressed the value of the present subgroup analysis among the Mediterranean countries of the HBSC in lines 84-88.

Point 3: Discussion, l.287-288: “The cluster analyses showed that the four separate clusters provide the most efficient CVD lifestyle risk factors profiling.” This statement is speculative, given that overall, only four CVD lifestyle risk factor were initially included in the data analysis without any forward or backward selection of a larger variable set.

Response 3: This sentence was rephrased in the following way: "Considering the four CVD lifestyle risk factors assessed, the best clustering solution which also yielded meaningful taxonomy of the data was four." (p. 11).

Point 4: Abstract: specify the number of modifiable CVD lifestyle characteristics that were examined: l. 33: “according to four modifiable lifestyle risk factors for CVD (…).”

Response 4: The four modifiable lifestyle risk factors are now listed in lines 31 and 35.

Point 5: Discussion, l: 291: “The prevalence of the five cardiovascular risk factors assessed.” Only four CVD risk factors were assessed – please correct.

Response 5: Corrected.  

Reviewer 2 Report

Summary

The authors did not specifically state the aim(s) of their study in the Introduction, but analyzed the profiles of adolescents from seven Mediterranean countries in relation to their modifiable risk factors for CVD. They identified four different cluster groups and also found that older adolescents reported a higher likelihood of being in the high-risk group. The conclusion can be improved upon.

Broad comments

Strengths: The article is well-written and flows logically between sections. Methods or procedures used are explained to give a reader context.

Specific comments:

Introduction:

  • Lines 55-56: Can authors give a more recent estimates than 2019? What were the rates in 2020 or 2021?
  • Lines 80-81: What is/are the aim(s) of the study? It was not clearly stated in the Introduction. What is the rationale for choosing these 7 specific geographical locations over any other locations?

Materials and Methods: 

  • Physical activity habits: Vigorous physical activity - Line 146: "... or sweet?" should read "... or sweat?"

Results

  • Table 1: Notes under table does not match table content in relation to change score and level of significance.

Discussion

  • Line 326: did you mean "prevention programmes aimed at CVD prevention"?
  • Lines 331-333: unclear sentence; please consider revising.
  • Line 339: "During adolescence, the drive for ..."
  • The authors did not discuss the implications of the difference in socio-demographic characteristics within the four clusters identified.

Conclusions

How will the identified clusters help in creating the tailored preventive interventions suggested by the authors?

Author Response

Response to Reviewer 2 Comments

Point 1: Summary

The authors did not specifically state the aim(s) of their study in the Introduction, but analyzed the profiles of adolescents from seven Mediterranean countries in relation to their modifiable risk factors for CVD. They identified four different cluster groups and also found that older adolescents reported a higher likelihood of being in the high-risk group. The conclusion can be improved upon.

Response 1: Thank you for this comment. The aim of the study was added to the end of the "Introduction" section on page 2. In addition, we added to the "conclusion" section on page 13 further information about the clusters. 

Point 2: Broad comments

Strengths: The article is well-written and flows logically between sections. Methods or procedures used are explained to give a reader context.

Response 2: Thank you for your comment

Point 3: Specific comments:

Introduction:

  • Lines 55-56: Can authors give a more recent estimates than 2019? What were the rates in 2020 or 2021?
  1. Response 3a. We replaced the reference from 2019 with one from 2021: World Health Organization. 2021. Cardiovascular diseases (CVDs): key facts. Available at:

https://www.who.int/en/news-room/fact-sheets/detail/cardiovascular-diseases-(cvds)

  • Lines 80-81: What is/are the aim(s) of the study? It was not clearly stated in the Introduction. What is the rationale for choosing these 7 specific geographical locations over any other locations?

Response 3b: The aim of the study was added to the end of the "Introduction" section (page 2).

The study focuses only on Mediterranean countries as the Mediterranean Region has many unique geographical, morphologic, historical, and societal characteristics related to CVD risk factors. For example, a growing body of evidence has shown that the traditional Mediterranean diet is associated with lower rates of both chronic diseases and premature death (Hutchins-Wiese, et al., 2021). This information was added to the end of the "Introduction" section on page 2.

Point 4: Materials and Methods: 

  • Physical activity habits: Vigorous physical activity - Line 146: "... or sweet?" should read "... or sweat?"

Response 4: Corrected.

Point 5: Results

  • Table 1: Notes under table does not match table content in relation to change score and level of significance.

Response 5: Our sincere apologies. We did not find the problem in table 1. We reviewed all the tables and did not find a mismatch between the table content and notes under the tables. We will be happy to correct the problem if you will provide us with a more detailed explanation.

Point 6: Discussion

  • Line 326: did you mean "prevention programmes aimed at CVD prevention"?

Response 6a. We corrected it as follows: This study also has many strengths. For example, we address, for the first time, the prevalence of CVD risk factors among school children in Mediterranean countries. In public health, it may be beneficial for Mediterranean countries to collaborate to develop and implement CVD prevention programs. Identifying profiles of moderate and high-risk adolescents is crucial.

  • Lines 331-333: unclear sentence; please consider revising.

response 6b. Thanks for the suggestion. We removed it

  • Line 339: "During adolescence, the drive for ..."

Response 6C. We corrected: During adolescence, the drive for thrill-seeking behaviours is an integral part of the maturation process. However, this should not be used as an excuse to normalize such reckless behaviours [8,51].

  • The authors did not discuss the implications of the difference in socio-demographic characteristics within the four clusters identified.

Point 6d. We added to the "Discussion" section the following paragraph, which discusses the differences in socio-demographic characteristics within the four identified clusters (pages 12-13):

"In the study, older adolescents and those from lower socioeconomic status reported a higher likelihood of belonging to the high-risk group. The increased likelihood of being in the high-risk group among older participants was not surprising, as it is known that risky behaviors increase with age. For example, according to the Centers for Disease Control and Prevention (2017), the prevalence of drinking increased from 23.4% among 9th grade students to 42.4% among 12th grade students. Similarly, the prevalence of binge drinking was 10.4% among 9th grade students and 24.6% among 12th grade students. Moreover, physical activity, the most prevalent lifestyle risk factor, also tends to decrease with age. Accordingly, adolescents constitute one of the most sedentary populations [48]. Additionally, socioeconomic status plays an important role in adopting healthy lifestyle behaviors. For example, numerous studies have shown that adults from lower socioeconomic status are more likely to smoke. However, the literature addressing smoking among youth is inconsistent. Several studies have shown an association between smoking and socioeconomic status among youth in some European countries, while others have not uncovered such associations [49]. In the absence of scientific agreement regarding the role of socioeconomic status and healthy lifestyle behaviors, such as smoking, among youth, the current study indicates the need for specific policies and interventions for different target groups in order to increase healthy lifestyle behaviors associated with reduced CVD. "

Centers for Disease Control and Preservation (2017). Current and Binge Drinking Among High School Students — United States, 1991–2015. Available at: https://www.cdc.gov/mmwr/volumes/66/wr/mm6618a4.htm (accessed on 20 January 2022).

Sallis, J.F.; Bull, F.; Guthold, R.; Heath, G.W.; Inoue, S.; Kelly, P.; Oyeyemi, A.L.; Perez, L.G.; Richards, J.; Hallal P.C. Lancet Physical Activity Series 2 Executive Committee. Progress in physical activity over the Olympic quadrennium. The Lancet. 2016, 388(10051), 1325-36.

Liu, Y.; Wang, M.; Tynjälä, J.; Villberg, J.; Lv, Y.; Kannas, L. Socioeconomic differences in adolescents’ smoking: a comparison between Finland and Beijing, China. BMC Public Health. 2016, 16(1), 1-8.

Point 7: Conclusions

How will the identified clusters help in creating the tailored preventive interventions suggested by the authors?

Response 7: We elaborated on the topic of preventive interventions in the "Conclusion" section (page 14): "Given the growing rate of premature CVD-related deaths worldwide, public health authorities and governments should consider implementing policies, health promotion, and tailored preventive strategies that could help increase healthy lifestyle behaviors associated with reduced CVD. CVD-risk clustering is an important step in strategy development, as it may identify the most high-risk population for CVD. For example, the study's results show that youth from lower SES families are more likely to present CVD-risk factors, suggesting that the particular needs of this population should be addressed and programs developed to increase healthy lifestyle behaviors."
